# Computational Three-Dimensional Imaging System via Diffraction Grating Imaging with Multiple Wavelengths

**DOI:** 10.3390/s21206928

**Published:** 2021-10-19

**Authors:** Jae-Young Jang, Hoon Yoo

**Affiliations:** 1Department of Optometry, Eulji University, 553 Sanseong-daero, Sujeong-gu, Seongnam-si, Gyonggi-do 13135, Korea; kikijang@naver.com; 2Department of Intelligent IoT Engineering, Sangmyung University, 20 Hongjimoon-2gil, Jongno-gu, Seoul 03015, Korea

**Keywords:** 3-D computational reconstruction, image enhancement, diffraction grating imaging, multiple wavelengths

## Abstract

This paper describes a computational 3-D imaging system based on diffraction grating imaging with laser sources of multiple wavelengths. It was proven that a diffraction grating imaging system works well as a 3-D imaging system in our previous studies. The diffraction grating imaging system has advantages such as no spherical aberration and a low-cost system, compared with the well-known 3-D imaging systems based on a lens array or a camera array. However, a diffraction grating imaging system still suffers from noises, artifacts, and blurring due to the diffraction nature and illumination of single wavelength lasers. In this paper, we propose a diffraction grating imaging system with multiple wavelengths to overcome these problems. The proposed imaging system can produce multiple volumes through multiple laser illuminators with different wavelengths. Integration of these volumes can reduce noises, artifacts, and blurring in grating imaging since the original signals of 3-D objects inside these volumes are integrated by our computational reconstruction method. To apply the multiple wavelength system to a diffraction grating imaging system efficiently, we analyze the effects on the system parameters such as spatial periods and parallax angles for different wavelengths. A computational 3-D imaging system based on the analysis is proposed to enhance the image quality in diffraction grating imaging. Optical experiments with three-wavelength lasers are conducted to evaluate the proposed system. The results indicate that our diffraction grating imaging system is superior to the existing method.

## 1. Introduction

Three-dimensional (3-D) imaging plays a key role in 3-D techniques. Also, their application fields are very broad such as augmented reality, autonomous driving, entertainment, defense, and biomedical imaging [1,2,3,4,5,6]. The main components of a 3-D imaging system are divided into 3-D image acquisition, 3-D image processing, and 3-D visualization. Among them, integral imaging was proposed as one of the promising techniques for 3-D applications by Lippman in 1908 [1]. Normally, 3-D integral imaging systems employ a camera array, a lens array, or a moving camera to construct 3-D systems. They can be applied to various applications such as 3-D depth extraction, 3-D display, 3-D visualization, 3-D pattern recognition, 3-D reconstruction, etc. [7,8,9,10,11,12,13,14,15,16,17]. As the first step in integral imaging, 3-D image acquisition is essential in that its optical devices provide image data for 3-D objects as an image array. This image array is one of the effective 3-D image formats in 3-D image processing and visualization.

Recently, a diffraction grating imaging (DGI) system has been introduced to acquire and process a 3-D image [18,19,20,21,22]. The optical configuration of a DGI system consists of an amplitude diffraction grating in the form of a transmissive film, a camera, and a laser as a light source [18]. The optical structure of DGI is inexpensive and simple compared with the camera array system. In addition, the grating in use is thinner, lighter, and less expensive than a lens array. Thus, it is easy to concatenate two or more gratings to design a 3-D imaging system [19], Also, it is free from optical aberration that occurs in the lens array. Therefore, the diffraction grating imaging system can be one of the promising techniques in 3-D information processing systems such as 3-D object recognition and depth extraction.

A diffraction grating imaging system consists of two processes. One is an optical pickup process, using a diffraction grating. The other is a computational reconstruction process, producing a 3-D volume [20,21,22]. In the optical pickup process, 3-D objects are illuminated with a laser beam, and then the laser rays are scattered from the objects. The incident rays on the diffraction grating among the scattered rays are diffracted. Finally, a camera can capture these diffracted rays in the form of an image array that is called a parallax image array (PIA) in diffraction grating imaging [18,19]. In the computational reconstruction process, digital image processing based on the back-projection is employed to extract 3-D information from the parallax image array. Here, a suitable computational reconstruction technique for diffraction grating imaging is required since there is no pattern to distinguish between parallax images in the captured PIA [20].

However, the limited number of diffraction orders due to the diffraction efficiency limits the number of parallax images in DGI [18,19,21]. Also, the speckle noise of a laser source deteriorates the image quality in computational reconstruction [20]. Although some techniques were addressed to solve these problems [21,22], there are still many problems such as speckle noises from a laser source, artifacts from crosstalk between parallax images, and blurring from back-projection in computational reconstruction. Thus, it is required to continue researching new techniques to overcome these shortcomings and to improve the image quality in diffraction grating imaging.

In this paper, we propose a diffraction grating imaging system to reduce the problems of the system and to improve the image quality, employing multiple lasers with different wavelengths. In diffraction grating imaging, parallax images are periodically seen in space when we observe the rays scattered from the surface of an object through the diffraction grating. Here, the distance between each parallax image is defined as a spatial period. Since this period is proportional to the distance between the object and the diffraction grating, the depth of the object can be obtained from the space period. The spatial period is also proportional to the diffraction angle and the wavelength of the laser in use according to the diffraction grating equation. That is, the spatial period of the object image observed through the diffraction grating increases as the depth of the object increases and the wavelength increases. In addition, the parallax image arrays have different characteristics, depending on the nature of the wavelengths of the lasers. This means that the wavelength affects the image quality of the reconstructed 3-D image; thus, it is worth studying a diffraction grating imaging system, taking the multiple wavelengths of the lasers into account.

Figure 1 shows the proposed diffraction grating system. Our system consists of an optical pickup process and a corresponding computational reconstruction process. Three parallax image arrays are acquired via sequential illumination of lasers with three different wavelengths such as 635, 532, and 450 nm, respectively. Since the depth resolutions of those parallax image arrays are different from each other, a computational reconstruction method has to merge multiple volumes from multiple PIAs into a single volume. To do so, we introduce the analyses of the spatial period of each parallax image array, the depth resolution, and the parallax angle according to each wavelength, which are given as generalized formulas and data tables. Based on the analyses, we propose a 3-D computational reconstruction method where multiple 3-D images from the parallax image arrays are integrated into a 3-D image. To evaluate the proposed method, subjective evaluation is carried out since it is challenging to prepare the 3-D ground truth images for the resulting images from optical experiments in DGI to calculate a reasonable objective measure. Thus, optical experiments with lasers of three different wavelengths are conducted and the results are compared with the existing method subjectively [20].

## 2. Geometries of Diffraction Grating Imaging Considering Multiple Wavelengths

### 2.1. Basis of Diffraction Grating Imaging

For a beam incident normally on a grating with linear patterns, the intensity maxima occur at diffraction angles *θ_m_*, which satisfy the equation sin *θ_m_* = *m*λ/*a*, where *θ_m_* is the angle between the diffracted ray and the normal vector of the grating, and *a* is the distance between each slit or the grating pitch, and *m* is an integer representing the diffraction order. Since the diffraction angle is proportional to the wavelength according to the diffraction equation, the rays with different wavelengths in an incoherent beam are diffracted in different paths; and the diffracted patterns of the rays enable us to analyze the wavelength components of the incoherent beam. These diffraction grating properties apply to recent research to develop spectrometries and X-ray imaging systems [23,24,25,26,27,28].

On the other hand, while an object is illuminated with a plane laser of a single wavelength, the plane rays scattered from the surface of the object are observed through a transmission diffraction grating. It is observed that the rays containing surface information of the object are periodically imaged in space. Here, the rays of the object are observed as an image, which is called a parallax image [18,19,20]. In other words, the rays emanating from the surface of the object are diffracted by the grating. The diffracted rays for the object can be imaged in the form of a two-dimensional array and a captured version of those images is called a parallax image array (PIA). In addition, the distance between the periodically observed parallax images is defined as a spatial period [20]. Since the spatial period is proportional to the distance between the object and the diffraction grating, the spatial period is an important parameter providing depth information of a 3-D object. After that, a computational reconstruction algorithm generates a 3-D volume from a parallax image array using the spatial period [20].

Figure 2 shows basic geometries to explain the ray analysis in diffraction grating imaging, where a point object, its parallax images, and an imaging lens are introduced [20,21,22]. Here, a point object is located at (*x**_O_*, *y_O_*, *z_O_*) and a diffraction grating is located at a distance *d* away from an imaging lens in the *x*–*y* plane. The *z*-coordinates of all parallax images are the same as *z_O_* according to diffraction grating imaging. The grating order is limited in the first-order diffraction to understand the system easily without loss of generality. Let the −1st and 1st order parallax images be *PI*(*x*_−1st_, *y_O_*, *z_O_*) and *PI*(*x*_1st_, *y_O_*, *z_O_*) which are the ±1st order diffraction of the point object. Here, the point object at (*x_O_*, *y_O_*, *z_O_*) is the 0th order parallax image. The diffraction angle *θ* between the 1st order and the 0th order parallax images is written as *θ* = sin^−1^(*mλ_k_*/*a*), where *λ_k_* is the wavelength of the *k*th light source, and *a* is the grating pitch and *m* is the diffraction order such as 0, 1, or −1. The tangent of the diffraction angle provides the locations of the parallax images. Here are the *x*-coordinates as follows:(1)$$x_{mth}=x_{O}+\left|z_{O}-d\right|\tan\left(\sin^{-1}\left(\frac{m\lambda_{k}}{a}\right)\right)$$

The *y*-coordinates, *y_mth_* are straightforward by replacing *x_O_* with *y_O_* in Equation (1). Also, the spatial period *X*, the distance between parallax images *PI*s, is then rewritten by
(2)$$X=X_{0}=\left|z_{0}-d\right|\tan(\sin^{-1}\left(\lambda_{k}/a\right))$$

Then, the imaging points *I*(*x_mth_*, *y_nth_*, *z_O_*) on the pickup plane are calculated by the pinhole analysis in the form of
(3)$$I\left(x_{mth},y_{nth},z_{O}\right)=\left(\left(\frac{z_{I}}{z_{O}}\right)x_{mth},\left(\frac{z_{I}}{z_{O}}\right)y_{nth},z_{I}\right)$$
where *m* and *n* are the diffraction orders in the *x* and *y* coordinates, respectively, and the *z*-coordinate *z_I_* in the pick-up plane is determined by the Gaussian lens formula.

Figure 3 shows the geometries to understand the parallax angles for PIs. The real ray emanating from the point object passes through a point *G* on the diffraction grating. Also, it passes through the imaging lens center after diffraction and reaches the PI pickup plane. The ray arriving at *I*(*x*_1th_, *y_O_*, *z_O_*) on the PI pickup plane looks as if it comes from the parallax image *P**I*(*x*_1th_, *y_O_*, *z_O_*). However, it emanates from the point object located at (*x_O_, y_O_*, *z_O_*). Thus, the diffraction at the point *G* on the grating enables the imaging lens to view the perspective scene of the point object, where the parallax angle *ϕ* of the parallax image *P**I*(*x*_1th_, *y_O_*, *z_O_*) can be expressed as the angle formed by the optical axis and the line segment connecting the point *G* and the point *PI*(*x_O_, y_O_*, *z_O_*). Here, based on triangular symmetry, the point *G*(*x_mth_*, *y_nth_*, *z_O_*) on the grating is written as
(4)$$G\left(x_{mth},y_{nth},z_{O}\right)=\left(\left(\frac{d}{z_{O}}\right)x_{mth},\;\left(\frac{d}{z_{O}}\right)y_{nth},\;d\right)$$

The parallax angle *ϕ_mth_* with a diffraction order *m* is then given by
(5)$$\phi_{mth}=\tan^{-1}\left(\left(\frac{d}{z_{O}}x_{mth}-x_{O}\right)\frac{1}{\left|z_{O}-d\right|}\right)$$

### 2.2. Depth Resolution and Spatial Period Depending on the Wavelength

An analysis of the diffraction grating imaging system concerning wavelengths is important in the proposed method since our diffraction grating imaging system employs multiple laser light sources with different wavelengths, which is called multiple wavelength diffraction grating imaging in this paper. Equation (2) says that the wavelength of the light source affects the spatial period in diffraction grating imaging. The depth information of an object is determined by the spatial period; thus, the spatial period can be considered as a function of the wavelength and object depth. Since the spatial period is a key parameter in the 3-D computational reconstruction process, it is necessary to calculate the spatial periods for one depth according to multiple wavelengths and it needs to integrate the 3-D images obtained from the parallax image arrays of multiple wavelengths. To solve this problem, an analysis of spatial periods is presented, depending on the wavelengths and depths.

Let us define the depth resolution as the minimum distance where two different depths can be recognized in the diffraction grating imaging system. As shown in Figure 2, the depth information *z_O_* of an object can be expressed as a spatial period *X*. Since the diffracted rays continue to travel straight, the diffraction angle *θ* is constant and its tangent can be calculated from a relation of tan(*θ*) = Δ*X*/Δ*z*. Thus, the depth resolution Δ*z* can be defined as the resolution Δ*X* of the spatial period and is given as
(6)$$\Delta z=\frac{\Delta X}{\tan(\sin^{-1}\left(\lambda_{k}/a\right))}$$

Here, if the resolution Δ*X* of the spatial period is set to the pixel size of the digital camera in use, the spatial period can be defined in a unit of a pixel number. Accordingly, the depth resolution is determined by Equation (6).

Figure 4 shows the relationship between depths and spatial periods in the parallax image arrays for multiple wavelengths utilized in the proposed diffraction grating imaging system, where the wavelengths *λ_k_* of the light sources are 635 nm (red), 532 nm (green), and 450 nm (blue), respectively. The parallax image arrays of a circular object are acquired using these three light sources, as shown in Figure 4a and their enlarged versions are shown in Figure 4b. Here, the circular object is located 100 mm away from the diffraction grating. As shown in Figure 4b, the spatial period of each parallax image array for the object depth was measured as 661, 553, and 463 pixels under the red, green, and blue light sources, respectively. It is seen that the larger the wavelength of a light source, the larger the spatial period. These results match well with Equation (2).

Figure 4c shows three graphs of space period vs depth for three wavelengths. It is seen that the graphs for the wavelengths are all linear. Here, the slope of each graph means the depth resolution and a constant slope means that the depth resolution is independent of the object depth. This property is an advantage compared to the fact that the depth resolution of a depth extraction system with a normal camera array deteriorates as the object distance increases. Also, it is seen that the slope increases as the wavelength increases, which means that the depth resolution is proportional to the wavelength. This result can be utilized as an important parameter in the design of depth extraction systems using diffraction grating imaging.

Moreover, the graphs of Figure 4c provide the key parameter in the 3-D computational reconstruction of a diffraction grating imaging system. Spatial period data for each wavelength need to be calculated to reconstruct an image at a target depth. They are essential to merge reconstructed images from multi-wavelength parallax image arrays. For example, as shown in Figure 4b, for a plane image reconstructed at a depth of 100 mm, the spatial periods should be set to 661 pixels for the red wavelength, 553 pixels for the green, and 463 pixels for the blue. Therefore, after a diffraction grating imaging system is designed and its graphs of Figure 4c are given, 3-D computational reconstruction is accomplished with those spatial periods for all wavelengths.

### 2.3. Parallax of PIA Depending on the Wavelength

This section describes parallax angles depending on wavelengths of light sources in our diffraction grating imaging system. The rays scattered from the surface of an object are diffracted at the grating and they are acquired by camera sensors. These captured rays are perceived as virtual images at different positions, thus a virtual image is considered as a version of viewing the object surface at a different angle. This angle is called the parallax angle in diffraction grating imaging. The geometries of a parallax angle of a parallax image are well described in Figure 3 and Equation (5). Based on this understanding, the effect of the wavelengths on parallax angles can be analyzed.

Figure 5 shows an example of three parallax images about three wavelengths in our diffraction grating imaging system and graphs of the parallax angles with an increase in distance. The optical configuration used to acquire the parallax image arrays is the same as that used in the previous section. A die of 3-D cube shape was selected as an object to present the effect of the wavelengths on parallax angles. The front face of the die is perpendicular to the optical axis and is located 100 mm away from the grating. The wavelengths *λ_k_* of the light sources in use are 635 nm (red), 532 nm (green), and 450 nm (blue). The parallax image arrays obtained from red, green, and blue wavelengths and their enlarged views of the parallax images of the first-order diffraction are shown in Figure 5a,b, respectively. As shown in Figure 5b, it is seen that the side of the die is observed as if it is viewed with a parallax angle.

Equation (5) says that the parallax angle is proportional to the spatial period. Since the spatial period is proportional to the wavelength, the larger the wavelength of the light source in use, the larger the parallax angle. This means that more 3-D information of an object can be obtained. This fact is well-demonstrated experimentally in Figure 5b. Based on this fact, three graphs of the theoretical parallax angles by the first-order diffraction with change in the object depth are depicted in Figure 5c. According to these graphs, it is seen that a parallax angle decreases slowly as the object moves away from the grating. Particularly noteworthy is the change in the parallax angle according to the wavelength, since the wavelength is an important optical design factor in the design of a diffraction grating imaging system.

## 3. Computational Reconstruction for Multi-Wavelengths Diffraction Grating Imaging

### 3.1. Virtual Pinholes and Mapping Positions of Parallax Images

Computational reconstruction in diffraction grating imaging with a single wavelength is based on the back-projection [20,21]. As discussed above, the parallax image array captured by our diffraction grating is a form of a 2-D image array, thus it looks like an image array obtained from a 2-D camera array. Since the image array from a camera array is easily utilized in back-projection techniques, it is required to understand the parallax image array theoretically as if a virtual camera array is engaged. Here, the virtual pinhole array model in DGI is a useful model to understand the virtual camera array [20]. Also, applying the virtual pinhole model to the proposed computational reconstruction is necessary to construct a multi-wavelength computational reconstruction method.

The ray coming from the point object at an angle of *ϕ* is related with the 1st order parallax image at (*x*_1st_, *y_O_*, *z_O_*), as shown in Figure 3. The ray going to the imaging point *I*(*x*_1st_, *y_O_*, *z_O_*) has the same angle of *ϕ*. The point *I*(*x*_1st_, *y_O_*, *z_O_*) is mapped with the parallax image of an angle of *ϕ*. As shown in Figure 6, let us draw a line passing through the point object and the point *G*(*x*_1st_, *y_O_*, *z_O_*) on the grating. This line meets the pickup plane. If taking another object point near the current object point, this line highlighted in blue also passes through the same point *VI*(*x*_1st_, *y_O_*, 0) on the *x*-axis. The point on the x-axis is called a virtual pinhole [20], as shown in Figure 6.

The virtual pinholes can be the optical centers and they are considered as an imaging lens array including the imaging lens. Equations (1) and (4) yield the position function *VP*(*x_mth_*, *y_nth_*, *z_O_*) in the form of
(7)$$VP\left(x_{mth},y_{nth},z_{O}\right)=\left(\frac{\left(x_{mth}-x_{O}\right)d}{z_{O}-d},\frac{\left(y_{nth}-y_{O}\right)d}{z_{O}-d},0\right)$$
where *m* and *n* are diffraction orders, and *x_mth_* and *y_nth_* are the *x*-position for order *m* and the *y*-position for order *n*, respectively. Here, the 0th virtual pinhole is the imaging lens. Equation (7) says that the position of each virtual pinhole increases in the *x* and *y* directions as the depth of the object increases. The virtual images are mapped from the images *I*(*x_mth_*, *y_nth_*, *z_O_*) in the pickup plane, and their x-coordinate is given by
(8)$$x_{VImth}=\left(\frac{x_{mth}d}{z_{O}}-x_{O}\right)\frac{z_{O}-z_{I}}{z_{O}-d}+x_{O}$$ where *y*-coordinates *y_O_* of these virtual images are straightforward by replacing *x_O_* and *x_mth_* with *y_O_* and *y_mth_* in Equation (8). The virtual images *VI*(*x_mth_*, *y_nth_*, *z_O_*) can be utilized in the computation reconstruction method of diffraction grating imaging. These images are also a shift version of the picked-up parallax images *I*(*x_mth_*, *y_nth_*, *z_O_*) with a factor of Δ*x_mapping_*. The shift Δ*x_mapping_* between *I*(*x_mth_*, *y_nth_*, *z_O_*) and *VI*(*x_mth_*, *y_nth_*, *z_O_*) from Equations (3) and (8) is written as
(9)$$\Delta x_{mapping}=\left|\frac{z_{I}-d}{z_{O}-d}\left(x_{O}-x_{mth}\right)\right|$$

### 3.2. Proposed Multi-Wavelength Computational Reconstruction

A computational reconstruction method is required to produce a 3-D volume from a parallax image array in diffraction grating imaging since a parallax image array of diffraction grating imaging has a different form compared with those of a lens array or camera array. For this purpose, a computational reconstruction method suitable for diffraction grating imaging was proposed in our previous research [15,20]. However, the previous methods were developed for a single wavelength. In this paper, a 3-D volume reconstruction method with multiple wavelength parallax image arrays is proposed to improve the image quality of diffraction grating imaging.

Figure 7 shows the computational reconstruction method for the proposed diffraction grating imaging system with multiple wavelengths. Here, the objects are sequentially illuminated with lasers of multiple wavelengths. The parallax image arrays of the objects are captured by our pickup process and apply to the back-projection process as input data. The back-projection based on the virtual pinhole array provides a reconstructed plane image at a given depth *z* for each wavelength. Here, the wavelengths *λ_k_* of the light sources in use are 635 nm (red), 532 nm (green), and 450 nm (blue), thus three reconstructed images are produced in this case. Those reconstructed images from multiple wavelengths must be at the same depth, which is guaranteed by the spatial period calculation method of the graph in Figure 4c. Then, reconstructed images corresponding to their single wavelength are applied to a spectral selection method. After the integration or combination process, the final reconstructed plane image is obtained at the given depth *z*. Repeating this process for all depths provides a 3-D volume in diffraction grating imaging.

## 4. Optical Experiments and Discussions

Figure 8 shows an optical experimental setup to evaluate the proposed 3-D imaging system via multiple wavelength diffraction grating imaging. As shown in Figure 8a, the experimental objects were placed 100 mm away from a diffraction grating, and the diffraction grating was located at 300 mm from a camera. The diffraction grating in use is made of two transmissive amplitude diffraction gratings with a line density of 500 lines/mm that are available commercially through perpendicularly concatenating two diffraction gratings in our laboratory. Three types of lasers with an optical power of 4.5 mW and the wavelengths of 635 nm (red), 532 nm (green), and 450 nm (blue) are utilized to illuminate the objects sequentially. Then, parallax images observed through the diffraction grating are acquired by a digital camera with a CMOS sensor of 35.9 × 24 mm and a pixel pitch of 5.95 μm to obtain parallax image arrays. The digital image size of each parallax image array is 3007 × 3007 pixels. Examples of three parallax image arrays corresponding to three types of wavelengths are shown in Figure 8b.

Figure 9 shows front and side views of the objects used in our optical experiments, parallax image arrays acquired by the pickup process, and their enlarged parallax images marked with a red square. As shown in Figure 9a, three flat objects of ‘S’, ‘M’, and ‘U’ were employed for optical experiments, and their positions were 100, 110, and 120 mm from the diffraction grating. In addition, as shown in Figure 9b, two human miniatures were used, and the 3-D objects are located 100 and 110 mm away from the diffraction grating. Other physical data regarding these objects are shown in the first column of Figure 9. Observing the enlarged parallax images in Figure 9, it is seen that the size of objects located at the same distance is the same regardless of the wavelengths of the light sources. On the other hand, the images from the red wavelength are rather blurred compared to those from the green and blue wavelengths. For example, as shown in Figure 9b, the buttons in the front human object are invisible at the red wavelength whereas they are visible at the green and blue wavelength. The random noises arising from the laser speckle noise are much stronger at the green and blue wavelengths. In addition, crosstalk between parallax images is seen in the parallax image from the green wavelength. The green horizontal bar in the enlarged part of the green image in Figure 9b is the crosstalk that arises from overlapping between parallax images due to the nature of diffraction grating imaging. Also, uneven intensity occurs in the enlarged part from the blue wavelength in Figure 9b. This uneven illumination is stronger as the wavelength is shorter in our experiments. Therefore, the parallax images suffer from various problems such as speckle noises, blurring, crosstalk, and uneven illumination.

Figure 10 presents the 3-D images reconstructed from three parallax image arrays in Figure 9a. Here, the row of reconstructed images indicates the depths of reconstructed images, and the column indicates the wavelengths and their combinations. The three columns from the left show the reconstructed images obtained from three single-wavelength parallax image arrays. The four columns from the right show the reconstructed images using multi-wavelength parallax image arrays. Particularly, the last column shows the 3-D images reconstructed from three wavelengths. The reconstructed images in each column of Figure 10 are expressed in colors according to each wavelength or its combinations. For example, a combination of the red and green wavelengths provides reconstructed images in yellow. Here, the reconstructed images from the green wavelength are the same as those from the existing method [20], where the green wavelength only is utilized. These optical experiments provide the differences in the image quality at different wavelengths. Thus, blurring or noises in reconstructed images can be different at different wavelengths and their combinations. Therefore, these results indicate the image quality is affected by the wavelengths of the illuminators in DGI and the wavelength of illumination can be an important parameter to design a DGI system. A more detailed discussion on these will be in the next paragraph with Figure 11.

Figure 11 shows the enlarged images reconstructed at a depth of 110 mm for the object ‘M’ in Figure 10 to compare and evaluate the reconstructed image quality. Three images in Figure 11a–c are reconstructed from the parallax image arrays at three single wavelengths, respectively. Among those images, the image from the red wavelength suffers from blurring whereas the blue wavelength has much less blurring. However, occluding noise arising from the object ‘S’ in the back-projection process exists in the image from the blue wavelength. This problem can increase as the spatial period decreases. Also, it is the same as the wavelength and the number of parallax images. It means that there are advantages and disadvantages according to the wavelength. Four images in Figure 11d–g are reconstructed from multiple-wavelength parallax image arrays. For example, as shown in Figure 11g, the image from three wavelengths suffers from blurring less than those images from the red and green wavelengths. Also, it has fewer occluding noises than the image from the blue wavelength. The background noises in Figure 11g are substantially reduced in the reconstructed image whereas the noises are more visible in the images from single wavelengths, for example, it is seen that the object ‘S’ is visible in the images from single wavelengths. The background noises are related to the number of parallax images, that is, the noises decrease as the number increases. Here, the parallax image number from multiple wavelengths is two or three times larger than the number from a single wavelength. Therefore, the image quality and resolution of the 3-D images of the proposed diffraction grating imaging system with multi-wavelengths are significantly improved, compared to the existing method [20].

Figure 12 presents the 3-D reconstructed images from the parallax image arrays in Figure 9b. Here, the meanings of the row and column in Figure 12 are the same as those in Figure 10. The reconstruction depth, however, was set to an arbitrary position since the shapes of the three-dimensional objects are arbitrary. The images in the three left columns in Figure 12 are reconstructed from single wavelengths. The images in the four right columns are from multiple wavelengths. It is seen that 3-D imaging works for all wavelengths since the object surfaces are seen at those depths. The images are affected by each wavelength and combination, similar to that of the images in Figure 10. In this experiment, the crosstalk between parallax images can be discussed. The images from the red wavelength are blurred without the crosstalk problem. However, the images from the green and blue wavelengths suffer from the crosstalk, for example, a green horizontal bar at the upper half of the images from the green wavelength is the crosstalk. Regarding this crosstalk, a detailed discussion will be in the next paragraph with Figure 13.

Figure 13 shows enlarged versions of the reconstructed images of a depth of 100 mm for the human object in Figure 12 to evaluate the image quality. Three images are reconstructed from single wavelengths, as shown in Figure 13a–c. Among these images, the image from the red wavelength suffers from strong blurring whereas the image from the green wavelength is much less blurred than that of the red. However, the image from the green wavelength suffers from the crosstalk, where the green horizontal bar in the image is the crosstalk from another parallax image. These unwanted artifacts exist in the image from the blue wavelength, as indicated by yellow and red arrows in Figure 13. This artifact arises from the interference or overlapping between the parallax images, which is unavoidable due to the nature of diffraction grating imaging. Such crosstalk artifact is more likely to occur as the spatial period is shorter or the illumination is stronger. For example, the image of the red wavelength unlikely occurs in the crosstalk, as shown in Figure 13a, whereas the image of the green wavelength with strong illumination can suffer from the crosstalk severely. On the other hand, in the images from multiple wavelengths, as shown in Figure 13d–g, the crosstalk is suppressed in the reconstructed images from multiple wavelengths. For example, the image from the red and blue wavelength, as shown in Figure 13c shows the best performance in terms of the crosstalk problem. Also, the crosstalk is substantially reduced in the image from three wavelengths. Therefore, the proposed diffraction grating imaging system with multiple wavelengths is more robust against the crosstalk problem than the system with a single wavelength.

Figure 14 shows the reconstructed images of a depth of 104 mm and their enlarged portions of the areas indicated by the yellow rectangles. As shown in Figure 14a–g, the reconstructed images are compared in terms of the sharpness of the surface of a human object. Particularly, the area indicated by arrows reveals the blue wavelength and provides higher image quality than the red and green wavelengths while the speckle noise remains stronger. However, the proposed computational reconstruction method with multiple wavelengths provides higher resolution images than the method with a single wavelength by reducing the speckle noise, the blurring, and the crosstalk, as discussed above. For example, the image from three wavelengths suffers from much lower speckle noises and crosstalks than a single wavelength.

Figure 15 shows the reconstructed images at a depth of 108 mm and their enlarged portions of the areas indicated by the yellow rectangles. As shown in Figure 15a–g, the areas are well focused in all reconstructed images, thus sharp edges are visible in the areas. The selected area of the human object is homogenous, thus it is useful to compare those reconstructed images in terms of smoothness. The image from the red wavelength, as shown in Figure 15a, has a homogenous area with sharp edges. This is useful to extract a homogenous depth map. However, the images from the green and blue wavelengths suffer from impulsive noises. Also, they have shadow areas that are unwanted signals in a depth extraction method. On the other hand, the images from multiple wavelengths have sharper edges than those from single wavelengths, observing the area indicated by arrows. Especially, the image from three wavelengths has a homogenous area with less impulsive noises and shadow areas.

## 5. Conclusions

This paper has presented a computational 3-D imaging method using diffraction grating imaging with multiple wavelengths. It is important to take account of the key parameters such as parallax angles, spatial periods, and depth resolution in the overall system performance when we design a 3-D imaging system via diffraction gratings. In particular, our study has shown that the wavelength of the light source can have a profound effect on these parameters. Optical experiments with three-wavelength lasers have been conducted to verify the proposed diffraction grating imaging system with multiple wavelengths. The results have indicated that the proposed diffraction grating imaging system is superior to the existing method with a single wavelength. Therefore, we expect that our DGI system will apply to 3-D imaging systems including depth extraction of 3-D objects that will be one of our future works.

## Figures and Tables

**Figure 1 sensors-21-06928-f001:**
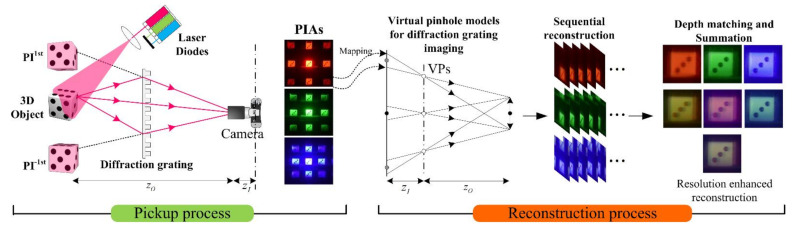
The proposed diffraction grating imaging system. It consists of a pickup process to acquire multiple PIAs from multiple wavelengths and a computational reconstruction process to generate 3-D images.

**Figure 2 sensors-21-06928-f002:**
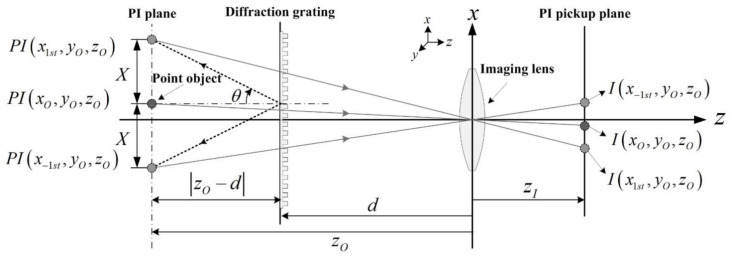
Geometries in diffraction grating imaging among a point object, parallax images (PIs), diffraction grating, an imaging lens, and captured parallax images.

**Figure 3 sensors-21-06928-f003:**
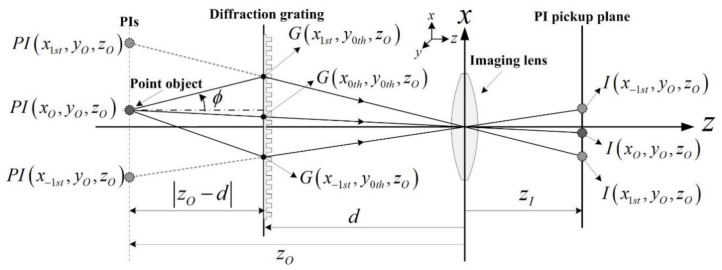
Geometries for the parallax angles of PIs in diffraction grating imaging.

**Figure 4 sensors-21-06928-f004:**
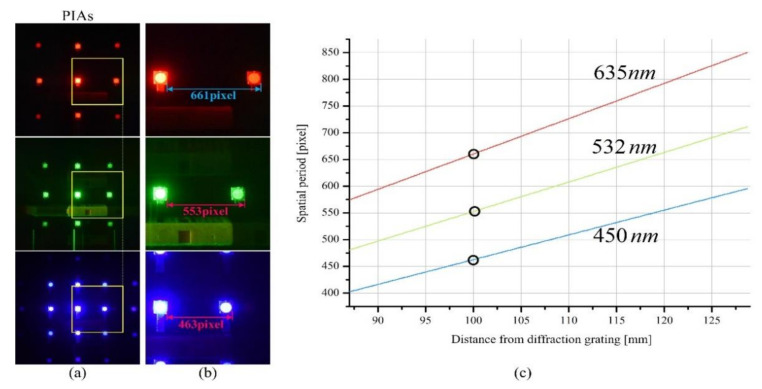
Spatial periods in pixels vs depth and wavelengths. (**a**) Parallax image arrays (PIAs) of wavelengths of red (635 nm), green (532 nm), and blue (450 nm). (**b**) Enlarged views for measuring three spatial periods. (**c**) Three spatial period graphs according to object depths.

**Figure 5 sensors-21-06928-f005:**
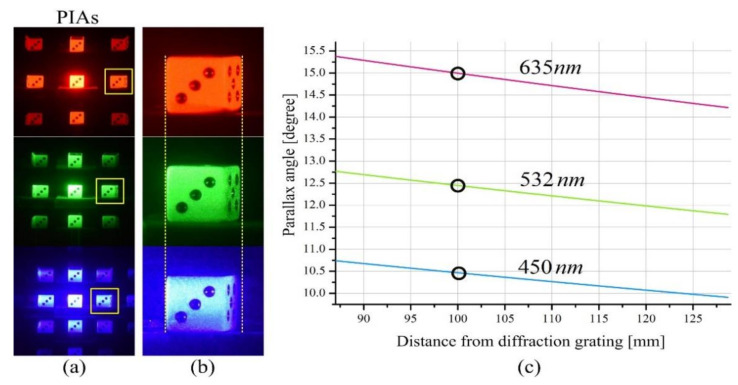
Parallax angles vs depth and wavelengths. (**a**) Parallax image arrays (PIAs) of wavelengths of red (635 nm), green (532 nm), and blue (450 nm). (**b**) Enlarged views of the parallax images by the 1st order diffraction. (**c**) Parallax angle graphs for each wavelength with an increase in the object depth.

**Figure 6 sensors-21-06928-f006:**
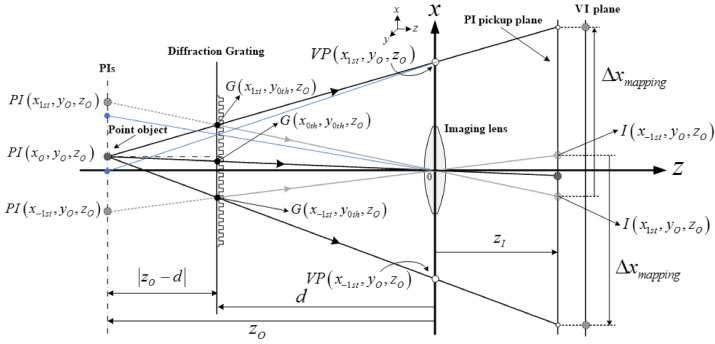
Geometries for virtual pinhole model among a point object, virtual pinholes (VP), virtual image (VI) plane, and *I*(*x*_1st_, *y_O_,z_O_*). For more details see text.

**Figure 7 sensors-21-06928-f007:**
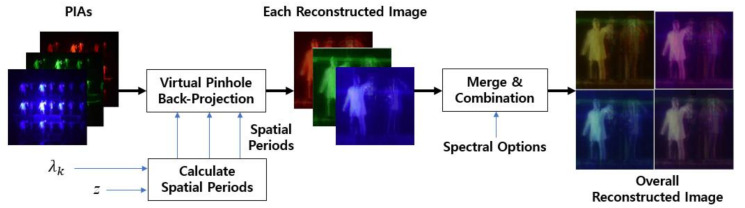
Proposed multi-wavelength computational reconstruction.

**Figure 8 sensors-21-06928-f008:**
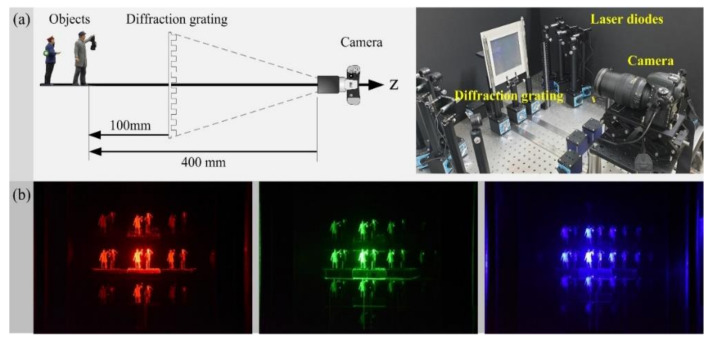
Experimental setup for PIA pickup. (**a**) Configuration of the optical experiment. (**b**) Examples of PIAs obtained from using lasers of wavelengths of 635 nm (red), 532 nm (green), and 450 nm (blue). See the text for the details on the configuration of optical devices.

**Figure 9 sensors-21-06928-f009:**
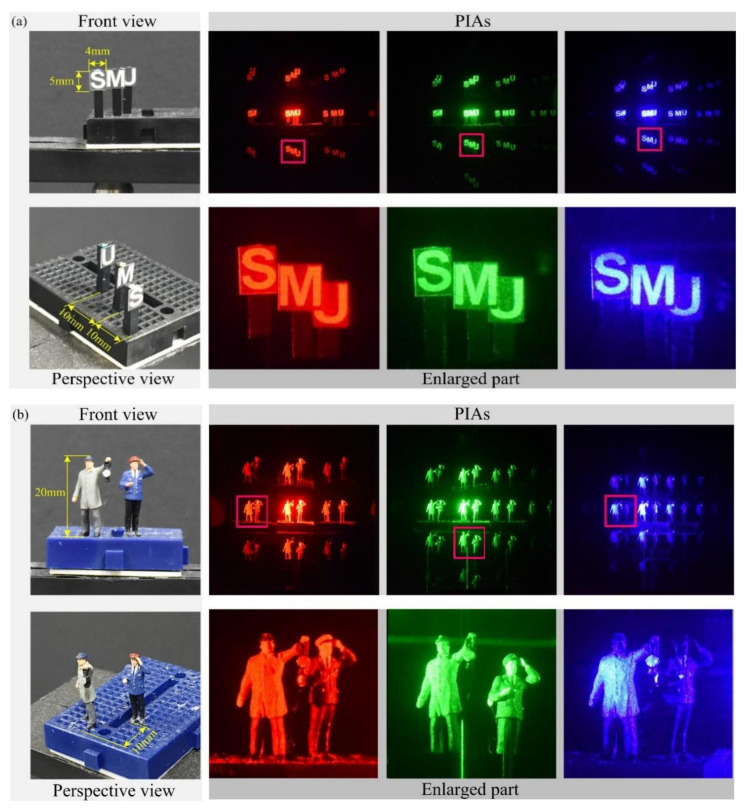
Objects used in our optical experiments and their PIAs and examples of a parallax image of the first-order diffraction in each PIA for (**a**) three flat objects of ‘S’, ‘M’, and ‘U’ and (**b**) two human miniatures. See the text for the discussion on the captured PIA.

**Figure 10 sensors-21-06928-f010:**
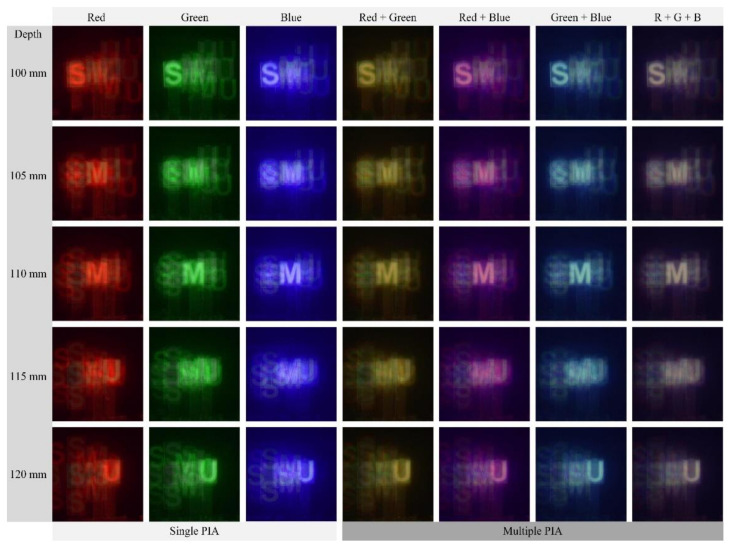
3-D reconstructed images for the PIAs in Figure 9a. Each image of three columns’ left is reconstructed from a single PIA. Each image of four columns’ right is reconstructed from multiple PIAs. The row indicates the depth of each image with a step of 5 mm.

**Figure 11 sensors-21-06928-f011:**
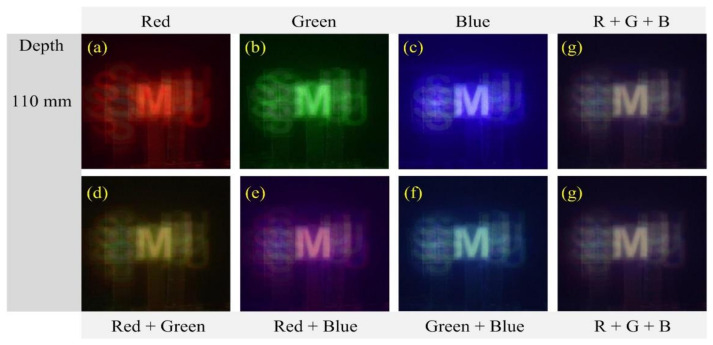
Reconstructed images for the object ‘M’ at a depth z = 110 mm. (**a**) Red (**b**) Green (**c**) Blue (**d**) Red + Green (**e**) Red + Blue (**f**) Green + Blue, and (**g**) Red + Green + Blue.

**Figure 12 sensors-21-06928-f012:**
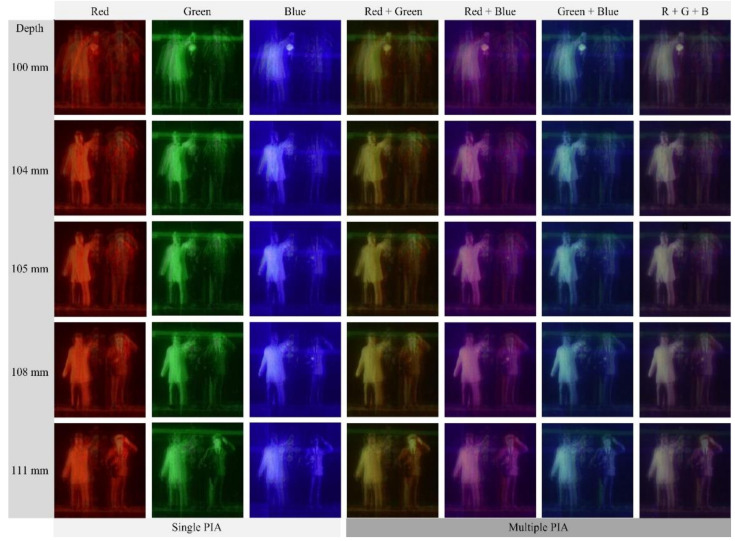
3-D reconstructed images for the PIAs in Figure 9b. Each image of three columns’ left is reconstructed from a single PIA. Each image of four columns’ right is reconstructed from multiple PIAs. The row indicates the arbitrary depth of each image.

**Figure 13 sensors-21-06928-f013:**
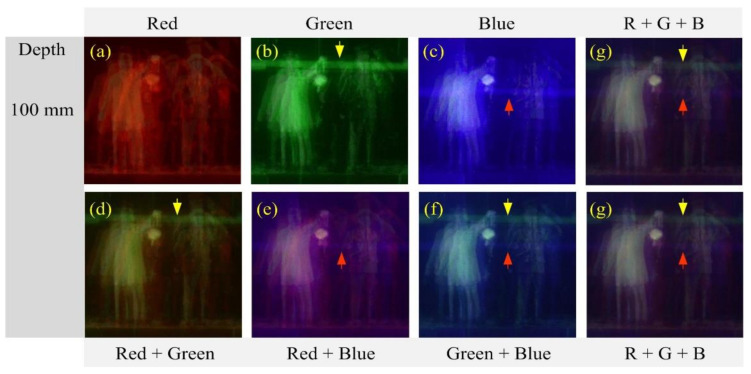
Reconstructed images for the human object at the depth z = 100 mm. (**a**) Red (**b**) Green (**c**) Blue (**d**) Red + Green (**e**) Red + Blue (**f**) Green + Blue, and (**g**) Red + Green + Blue wavelength.

**Figure 14 sensors-21-06928-f014:**
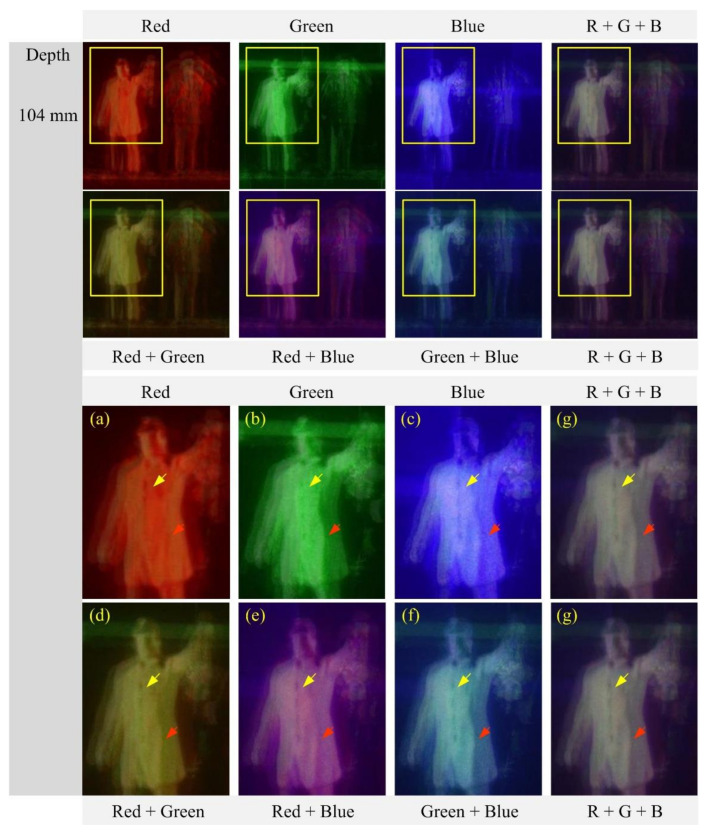
Reconstructed images and their zoomed versions for the human object at the depth z = 104 mm. (**a**) Red (**b**) Green (**c**) Blue (**d**) Red + Green (**e**) Red + Blue (**f**) Green + Blue, and (**g**) Red + Green + Blue wavelength.

**Figure 15 sensors-21-06928-f015:**
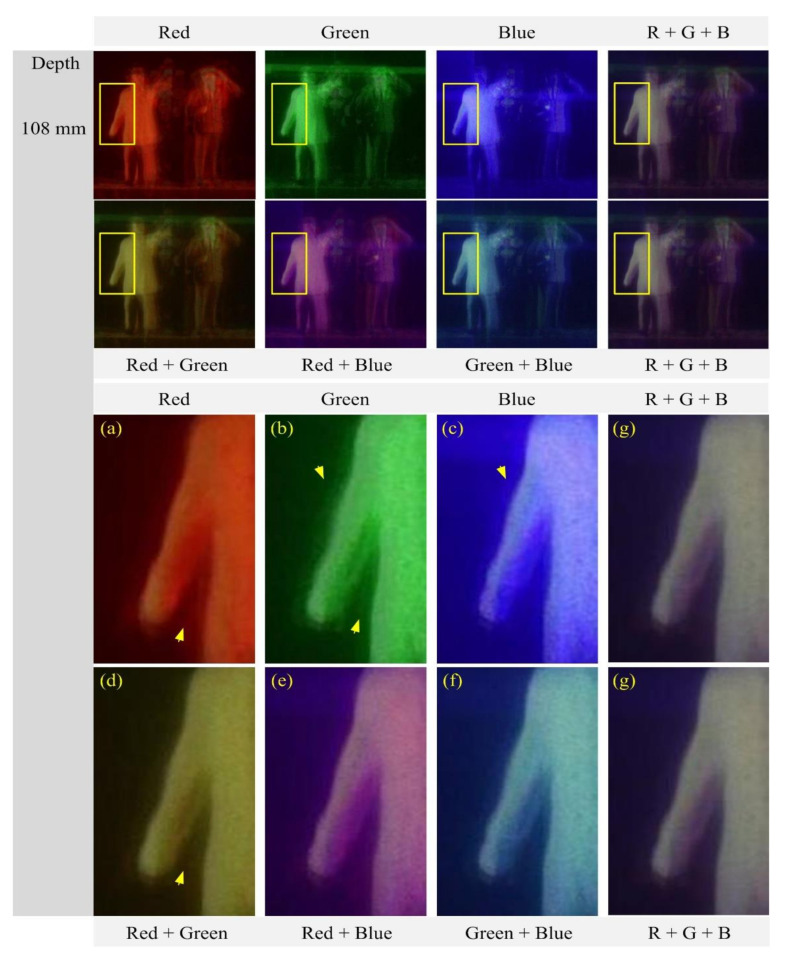
Reconstructed images and their zoomed versions for the human object at the depth z = 108 mm. (**a**) Red (**b**) Green (**c**) Blue (**d**) Red + Green (**e**) Red + Blue (**f**) Green + Blue, and (**g**) Red + Green + Blue wavelength.

## Data Availability

Data sharing not applicable.

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
