# Peer review of "Computational Three-Dimensional Imaging System via Diffraction Grating Imaging with Multiple Wavelengths"

_sensors, 2021, doi:10.3390/s21206928_

Round 1

Reviewer 1 Report

     The diffraction grating imaging system suffers from noises, artifacts, and blurring due to single-wavelength lasers' diffraction nature and illumination. This paper describes a computational 3-D imaging system based on diffraction grating imaging with laser sources of multiple wavelengths. The results indicate that the proposed diffraction grating imaging system is superior to the existing method.

     The structure of the manuscript is well. The manuscript has some of its technical merits. The topic is very interesting. It also falls into Journal Sensors. However, several questions may need to pay attention to:

  1. The English language is not good enough for the acceptance of the International Journal. There are some typos in the manuscript. Recommend the authors should check the manuscript carefully before submission.
  2. In section 1, the authors give the abbreviation of the diffraction grating imaging system at the first time in line 45, not other locations in the text.
  3. In section 1, the authors proposed the question in diffraction grating imaging. However, the issue's significance has been described much more in the paper. Recommend the authors should emphasize it.
  4. The caption of Figure 1 more likes the description of the content in detail. The detailed description can be moved to text. The caption needs to be improved and shorted.
  5. Some images in the Figure display in low-resolution pixels, such as Figure 1-3. Recommend the authors should use high-resolution pixel images.
  6. In equations, each variable has its annotation. The authors need to give the annotation and necessary explanation.
  7. There is one significant problem with how to measure the performance of the proposed system. The authors only provide the image but not performance indices, such as SNR…. The problem needs to be addressed.
  8. In section 4, the authors give some description with the images but no explanation. Recommend the author should add some explanation.
  9. The discussion can be improved. The authors have not pointed out the disadvantage of the proposed algorithm or comparison of enough previous literature. Recommend the author should extend your discussion.
  10. In conclusion, the authors might suggest future research. It will be better.

Hopefully, this will help in the revision of the manuscript.

Author Response

Please see the file attached. 

Reviewer 2 Report

The paper presents an imaging system capable to sense the field depth (ie 3D) by imaging the scene through a diffraction grating under the illumination of different wavelengths of light. Although the performance of the present design may not be as good as that of the light field imaging system, the very simple and low cost of the system makes the idea rather interesting and it is worth for publication. However, there are many problems with this paper and they are needed to address before it can be accepted for publication.

  1. This paper is an extension of the authors’ previous publications (see ref 18-22) by using multiple wavelength of light as illumination source for enhancing the quality of the depth image, which can be considered as the new concept and is worth to be published. However, the paper gives a very detailed explanation of the 3D imaging by using diffraction grating (see sections 1-2.1) which should be shortened and summarised substantially.
  2. One of the greatest drawback in this paper is the very poor use of English throughout the paper, which makes the reader very hard to follow what the authors are trying to illustrate. This will give very negative impact to the paper and the reputation of the journal. The paper is needed to be completely rewritten to let the reader to understand, and, to appreciate, the innovation of the paper.
  3. The abstract should be rewritten as it cannot convey the innovation of the paper at all.
  4. Are the line plots in Fig 4c and 5c are the experimental data? If so can the data points be plotted too?
  5. The reviewer is not able to make further comments as the paper is so difficult to follow. Please rewrite the paper and resubmit.

Author Response

Please see the file attached. 

Reviewer 3 Report

Overall

The work reported in this submission concerns the improvement of a previously published 3-D Diffraction Grating Imaging system (DGIS) that was found to have advantages over established systems that use lens arrays or camera arrays. The authors rightly point out a number of problems that exist with DGIS e.g. noise and blurring linked to aspects of single wavelength lasers. The influences of multiple lasers was therefore examined, viz spatial periods and parallax angles. The development of a computational 3-D imaging system, with 3 wavelengths, sought to improve image quality of the DGIS and this was to some extent achieved. It was concluded that the authors’ DGIS was superior to other existing methods.

This paper represents a reasonable piece of work in this rather specialised field. The comparisons between the DGIS and the previous or standard systems is qualitative. This makes it difficult to give definitive results especially as the differences in image quality, noise, and artefact are not that significant. There are some relatively minor revisions/corrections/re-wording that would improve the paper a lot in terms of understanding by readers of this journal. The text needs to be reviewed thoroughly to improve style, punctuation, and English

Abstract:

Line 12. “…in the previous studies…”. Are these studies by the authors or by other researchers? Say which. Also it claims that a DGIS is superior to the systems based on a lens array etc., but at the same time further work is needed on the DGIS to improve noise etc.; this sounds equivocal. Why should a multiple wavelength system overcome these issue

1. Introduction

a. Integral imaging is referred to (lines 31 and 34) and some 15 references to it are given (page 17), but the introduction lacks a simple description of what “integral imaging” is in practice.

b. The 3-D image acquisition is said to be important, so what features of the systems cited in refs [7-17] present the most important problems or limitations to be overcome? It could be made clearer that refs [18-22] are the work of the authors, in which their novel contributions to the field of Diffraction Grating Imaging are published.

c. Presumably, the actual DGIS constructed by the authors (lines 40-48 and Fig 1)) is the only one of its kind; please ensure that any other systems or parts thereof are cited where appropriate. Also, image reconstruction via back projection algorithms is the subject of a large literature, including the applications mentioned by the authors. So, please include a reference to the back projection regime that is currently favoured in the diffraction grating imaging field.

d. The explanation given for the influence of the laser wavelength on image quality is reasonable. Hence the intention to assess the performance when multiple wavelengths are used is valid. However, the statement of the main aim of the research (lines 93-99) here does not make it clear just how a comparison with “…the existing method….” will be carried out

2. Geometries of diffraction grating imaging….

e. The description of sub-section 2.1, which is the basis of diffraction grating imaging, is sound. It is mentioned (line 111) that the diffraction grating properties can apply to spectroscopy; what is the significance of this? The definitions for ‘parallax image’ and ‘parallax image array’, (lines 116, 120), also referred to in the authors’ earlier publications, are appropriate.

f. In Figure 3, the angle Ψ does not appear to be defined; does it relate to the discussion of the parallax angles

3. Computational reconstruction for multi-wavelengths

g. The need for using a virtual pinhole model is clearly justified in the text.

h. Something missing on line 289 ?

A general comment for the Methodology, the Results, and the Discussion is that these important sections are rather poorly constructed/worded. In some parts the text is difficult to follow and in other instances some details are missing. In fact, the paper does not have a distinct section for Methodology, although some details are given in section 4, lines 343-352. The paper could be improved by: giving details of the laser diodes used; giving details of the camera and justifying the use of this particular device (line 346); Figure 6 is very well drawn and detailed, but the captions for Figures 6, 8 and 9 need to have more details, even if the captions state “for more details see text”; It is not clear if the diffraction gratings were lab made or commercial products – clarify and give further details.

4. Optical experiments and discussions

[See 3i above]

j. In Figure 8 it is not clear why the figure 8 (a) is essentially repeated in the lower part of figure 8 (b); please clarify. Also the three flat objects are “S, M, and U” not “S, N and U” (line 367). The statements given in lines 372 to 376 are not easily seen in the enlarged parts of figures 9 (a) and 9 (b), as differences in noise and blurring are difficult to see. Please attempt to state these criteria (noise, artefact, and blurring) as quantitative measures. This is necessary in order that these performance characteristics for the “…existing method…” can be compared with those achieved with the multi-wavelength system.

k. The results presented in Figure 10, using the 3-D reconstructed images from Figure 9, show single wavelength PIAs alongside combined multiple PIAs. The latter are said to indicate enhanced reconstructed images arising from multiple wavelengths. The explanation of this is not clear. Perhaps there is a depth effect with which more blurring is seen progressively from 100 mm down to 120 mm, but this needs some discussion. Also for any given depth is it being claimed that there is more or less noise or artefact or blurring as one changes the combination of wavelength, say from (Red+Green) to (Red+Blue) ? All of this may be too subjective and this concern also applies to the reconstructed images presented in figure 11, and it is not clear that the stated crosstalk in figure 13 is clearly visible, linked to the red and yellow arrows. An explanation and discussion are needed for these results. However, in figure 14 the better image quality of the blue wavelength compared with the red wavelength is just discernible

l. In the reconstructed images, at a depth of 108 mm, with their zoomed versions in figure 15 the stated improvement with red compared with green and with blue is not clearly visible. Further consideration of this point is needed.

5. Conclusions

m. I suggest replacing the two sentences in lines 469-472 by: “When designing a 3-D imaging system based on such diffraction gratings it is important to take account of the influences that the key parameters of parallax angles, spatial periods, and depth resolution will have on the overall system performance. In particular, our study has shown that the wavelength of the light source can have a profound effect on these parameters.”

Author Response

Please see the file attached. 

Round 2

Reviewer 1 Report

The manuscript has been revised according to most of the points.

Author Response

We appreciate your kind recommendation.

Reviewer 2 Report

Technically the paper is sound but most merits of the paper have been stripped off by the rather unacceptable level of English which makes the paper very difficult to read. Although the authors have improved the English of the paper slightly, the clarity of the paper is still very poor due mainly to the poor use of English throughout the paper. There are so many broken and incomplete sentences that it simply put off readers not to read the paper in depth and fail to convey the technical innovation to the public as the result.

In my view the paper must be completely rewritten and resubmitted.  

Author Response

Question:

Technically the paper is sound but most merits of the paper have been stripped off by the rather unacceptable level of English which makes the paper very difficult to read. Although the authors have improved the English of the paper slightly, the clarity of the paper is still very poor due mainly to the poor use of English throughout the paper. There are so many broken and incomplete sentences that it simply put off readers not to read the paper in depth and fail to convey the technical innovation to the public as the result.

In my view the paper must be completely rewritten and resubmitted.  

Answer:

Thank you for your kind recommendation. We have revised the manuscript as the reviewer pointed out. The manuscript has been checked by a Canadian English speaker whose major is English education. Many parts of the manuscript are revised from the recommendation. We think the revised manuscript enables the reader to understand what is the proposed technique. Please see the revised manuscript which is highlighted in blue.

Reviewer 3 Report

The authors have considered and evaluated all of the comments made by this Reviewer. All of the points raised on the original manuscript have been dealt with appropriately and the paper has been improved as a result.

The authors have revised several parts of the paper and it will now be useful to have the manuscript checked once again for the English phrasing.

Author Response

Question:

The authors have considered and evaluated all of the comments made by this Reviewer. All of the points raised on the original manuscript have been dealt with appropriately and the paper has been improved as a result.

The authors have revised several parts of the paper and it will now be useful to have the manuscript checked once again for the English phrasing.

Answer:

Thank you for your kind recommendation. We have revised the manuscript as the reviewer pointed out. Many sentences are reworded in the manuscript. We think the revised manuscript enables the reader to understand what is the proposed technique. Please see the revised manuscript which is highlighted in blue.

Round 3

Reviewer 2 Report

The readability of the paper has been improved rather substantially although it is still not perfect, however, it is accepted as it is.